# Supporting families to protect child health: Parenting quality and household needs during the COVID-19 pandemic

**Leslie E. Roos**[1,2,3]*, **Marlee Salisbury**[1], **Lara Penner-Goeke**[1,4], **Emily E. Cameron**[1], **Jennifer L. P. Protudjer**[2,3], **Ryan Giuliano**[1,3], **Tracie O. Afifi**[1,5,6], **Kristin Reynolds**[1]

1 Department of Psychology, University of Manitoba, Winnipeg, Canada, 2 Department of Pediatrics and Child Health, University of Manitoba, Winnipeg, Canada, 3 Children's Hospital Research Institute of Manitoba, Winnipeg, Canada, 4 Department of Psychology, University of Winnipeg, Winnipeg, Canada, 5 Department of Community Health Sciences, University of Manitoba, Winnipeg, Canada, 6 Department of Psychiatry, University of Manitoba, Winnipeg, Canada

* leslie.roos@umanitoba.ca

**Data Availability Statement:** De-identified data relevant to this manuscript is available here:https://osf.io/yw8gr/.

## Abstract

### Background

Supportive parenting is critical for promoting healthy child development in the face of stressors, such as those occurring during COVID-19. Here, we address a knowledge gap regarding specific household risk factors associated with parenting quality during the pandemic and incorporate first-person accounts of family challenges and needs.

### Methods

Mixed methods were applied to data collected between April 14th - 28th, 2020 from the "Parenting During the Pandemic" survey. Participants included 656 primary caregivers (e.g., mothers, fathers, foster parents) of least one child age 1.5–8 years of which 555 (84.6%) responded to at least one parenting questionnaire. Parenting quality was assessed across stressful, negative, and positive parenting dimensions. Household risk was examined across pandemic- linked (e.g., caregiver depression, unmet childcare needs) and stable factors (i.e., annual income, mental illness history). Significant correlates were examined with regressions in Mplus. Thematic analysis identified caregiver challenges and unmet needs from open-ended questions.

### Findings

Caregiver depression, higher child parity, unmet childcare needs, and relationship distress predicted lower-quality parenting. Caregiver depression was the most significant predictor across every parenting dimension, with analyses indicating medium effect sizes, ds = .39 - .73. Qualitative findings highlighted severe strains on parent capacities including managing psychological distress, limited social supports, and too much unstructured time.

**Funding:** Initials of funded authors: LR; RG; JP; MS; LPG; KR Grants: 1) Research Manitoba - 2020 Manitoba COVID-19 Rapid Response 2) CHRIM - MG2020-3 - Match for project titled "The PACT Program: Parenting Apart while Coming Together Research Grant Funders Name: 1) The Children's Hospital Research Institute of Manitoba (CHRIM), Winnipeg, MB 2) Research Manitoba, Winnipeg, MB URL of Funders Website: https://researchmanitoba.ca/ https://www.chrim.ca/ The funders had no role in study design, data collection and analysis, decision to publish, or preparation of the manuscript.

**Competing interests:** The authors have declared that no competing interests exist.

## Interpretations

Lower quality parenting during COVID-19 is associated with multiple household and pandemic risk factors, with caregiver depression consistently linked to parent- child relationship disruptions. Focused efforts are needed to address caregiver mental health to protect child health as part of the pandemic response.

## Introduction

The COVID-19 pandemic has resulted in closures of daycares, schools, and recreational facilities across the world, affecting an estimated 1.38 billion children [1, 2]. Families are coping with numerous challenges including unmet childcare needs, crisis schooling, low resource access, and financial strain [3–5]. In such conditions, supportive parenting relationships are critical for preventing the effects of stress from affecting children's neurobiological and socioemotional development [6]. Exposure to early life stressors, including poverty, family conflict, and parent mental illness, can impair health across the lifespan [7]. Stress-exposed children are more likely to experience developmental delays, asthma, diabetes, obesity, mental illness, and hospitalization [8, 9]. Stress-linked social determinants of health compound impacts through increased health-risk behaviours [10]. These risks persist throughout life, contributing as much as a 4-fold increase in risk for physical and mental health conditions [11].

Notably, not all children exposed to early life stress go on to experience significant health risks. Supportive caregiving relationships are consistently identified as powerful protective factors for promoting resiliency to stress [12, 13]. Developmental origins of chronic disease research highlights both the protective neurobiological signaling and relational processes that underlie the significance of supportive parenting to promoting child resilience. At the neurobiological level, responsive caregiving to young children's distress supports physiological regulation and programs neurocircuitry underlying normative emotional development [14]. From a social-learning perspective, parents model stress-management and encourage different coping strategies over time. These strategies can include directly teaching young children how to handle strong emotions or responding in ways that increase or decrease different child attempts at coping over time, such as withdrawal or aggression.

Although the full scope of adversities experienced by children due to COVID-19 will be difficult to determine for many years, understanding the factors associated with low parenting quality during the current time is critical. Here, we provide observational mixed-methods data on sociodemographic- and pandemic-linked household stressors associated with parenting quality for 1.5- to 8-year-old children. We incorporate qualitative data on first-person parent perspectives to understand the unmet needs of caregivers during these unprecedented global events. The goal of this work is to inform next steps in identifying the specific strengths and needs of families during the pandemic and its aftermath to protect children from negative health consequences of pandemic-linked stress.

### Indices of parenting quality

**Parenting quality.** *Positive parenting* strategies include multiple dimensions of parenting linked to healthy child development including supportive parenting, setting limits, and proactive parenting [15]. Positive parenting involves a variety of behaviours, such as using praise, engaging in joint-play, setting effective routines, and showing warmth, which increase prosocial child behaviours [16]. Positive parenting decreases children's behavioural problems [16] and buffers children against the effects of chronic stressors, such as poverty and peer rejection

[17, 18]. *Negative parenting* strategies include over-reactivity, a tendency to respond to children's difficult behaviour with harshness and anger, and laxness, a tendency to ignore or not follow through with discipline for problematic behaviour [19]. Over-reactivity and laxness have been associated with young children's emerging mental health problems and are linked to stressors, such as economic strain [20, 21].

*Parent stress* includes distress related to the caregiving role as well as qualities of the parent-child relationship and perceptions of child difficulties [22]. High parenting stress has wide ranging consequences for child health, including increased risk of child abuse [23], obesity [24] and externalizing behavioural problems [25]. Greater parenting stress is also associated with more dysfunctional parent-child interactions including lower parenting sensitivity and higher prevalence of dysfunctional attachment styles [26, 27]. Within the neuroscience literature, research indicates increased stress may be associated with decreases in brain synchrony between mothers and their children [28].

**Pandemic- and sociodemographic-linked parenting risks.** Emerging data indicate multiple concerning pandemic-linked factors expected to limit parent capacities to provide high-quality caregiving, including job loss, food insecurity, domestic conflict, and parent psychological distress. In Canada and the United States, unemployment rates have increased 3- to 6-fold, with the global recession expected to contribute to ongoing financial insecurity. These economic stressors will compound existing inequities, with job losses most pronounced in low salary positions [29]. Such economic adversity has been associated with poor child health outcomes, which are consequently exacerbated by negative parent-child relationships and harsh parenting [30]. Food insecurity is expected to increase significantly due to economic strain, with vulnerable populations most affected [31]. Increased marital conflict and divorce has been reported following shelter-in-place restrictions in China, and increased domestic violence reports in North America [32, 33]. Data from our group and others indicate dramatic increases in parent psychological distress with 30–48% of mothers of young children reporting depression above clinical thresholds [34]. Each of these factors, alone and in combination, has been linked to lower parenting quality (e.g., low responsivity, increased harsh parenting, decreased positive parenting, disengaged parenting) and the emergence of child health and development impairments [35–42].

## The present study

The present study provides the first COVID-19 pandemic data on sociodemographic- and pandemic-linked risk factors associated with parenting quality. With an estimated one-third of the world's population under mandated lockdowns [43], understanding factors that impact a family's capacity to provide supportive caregiving is critical. This knowledge will allow for targeted intervention development to prevent long-term child health and developmental consequences in the aftermath of the pandemic. These data are provided to bring attention to this understudied and, as of yet, underfunded area of health investment need and may help identify which families are in most need of support to limit the negative health impacts of COVID-19.

Some of these risks may be most acute when families are isolated, and children do not have access to high quality daycare or school environments. However, other risks are expected to continue for months [44], due to the unpredictable nature of disease resurgence, childcare closures, and economic recession.

## Methods

### Participants

Between April 14, 2020 and April 28, 2020, 656 parents volunteered to complete an online survey on "Parenting During the Pandemic." Individuals were eligible for the present study if

they were above the age of 18 years and a caregiver of at least one child between the ages of 1.5–8 years old. This child age range was selected to be distinguished from the postpartum period, where parent-child interactions may differ from those during early childhood [45]. Early childhood is a sensitive period when children are highly reliant on primary caregivers for emotional and behavioural regulation and are most susceptible to environmental influence including the caregiving environment [46]. Moreover, descriptions of the psychometric properties and intended use of many parenting questionnaires begins at 18 months of age, thus the focus of the current study was on parenting during early childhood.

## Procedure

Participants were recruited online through social media platforms. Data collection occurred through the online health-information secure survey using REDCap (Research Electronic Data Capture) data capture tools hosted at the University of Manitoba [47]. Informed consent was obtained online prior to beginning the survey. Parents of multiple children identified their most-challenging child in the 1.5–8 year age range when completing the Parenting Stress Index and Parenting Questionnaire. Remuneration included a draw for one of five $100/CAD e-gift cards.

## Measures

**Stable household risk factors.** Sociodemographic and historical risk factors were examined as possible covariates. These included marital status (married or co-habitating versus single, widowed, or divorced), annual household income, caregiver education, number of children in the home and caregiver history of diagnosed mental illness (anxiety or depressive disorders).

**Current stressors related to COVID-19.** All stressors related to COVID-19 were assessed using a questionnaire created for this study. Employment loss was measured as any loss of hours including working reduced hours or being laid off. Financial strain was assessed as level of difficulty managing unexpected expenses. Food insecurity was assessed as any endorsement of insecurity, based on PROOF guidelines (i.e., at least one affirmative response indicating insecurity) [48].

**Revised Dyadic Adjustment Scale (RDAS).** Marital/relationship quality was assessed using the Revised Dyadic Adjustment Scale (RDAS), a 14-item self-report measure which yields subscales on three relationship domains: 1) Consensus, 2) Satisfaction, and 3) Cohesion [49]. The scale has acceptable internal consistency on subscale and total scale measures ($\alpha$ = .81-.90) and reliably distinguishes between distressed and non-distressed couples in both clinical and research settings [49].

**Parental mental health.** Mental health categorization was determined based on participants' scores on best-practice measures of anxiety [Generalized Anxiety Disorder 7-Item Scale (GAD-7) [50] or Perinatal Anxiety Screening Scale (PASS) [51]] and depression [Center for Epidemiologic Studies Depression (CESD) [52], CESD-Revised (CESD-R) [53], or the Edinburgh Postnatal Depression Scale (EPDS) [54] as well as EPDS-Partner [55]], based on child age and parent gender [34]. Clinical cut-off scores representing clinically significant anxiety and depression symptoms were used for categorization. Internal consistency of all parent mental health measures is high in previous literature (GAD-7: $\alpha$ = .92; PASS: $\alpha$ = .96; CESD: $\alpha$ = .85; CESD-R: $\alpha$ = .92; EPDS: $\alpha$ = .82) [50–54].

## Parenting

**Parenting Young Children (PARYC).** Positive parenting strategies were assessed using the Parenting Young Children (PARYC) scale, a 21-item self-report asking parents to rate how

often they engaged in specific parenting behaviours over the past month [15]. The PARYC yields subscales on three distinct areas of parenting: 1) Supporting Positive Behavior, 2) Setting Limits, and 3) Proactive Parenting [15]. Internal reliability for each subscale is acceptable ($\alpha$ = .78 - .85) [15].

**The Parenting Scale (PS).** Negative parenting strategies were assessed using the Parenting Scale (PS), a 30-item self-report measure of ineffective discipline strategies [19]. Parents are asked to rate the likelihood of using particular discipline strategies to yield two discipline subscales: Laxness and Over-reactivity [56]. The PS has good internal consistency ($\alpha$ = .63-.84), test-retest reliability ($r$ = .79-.84), and differentiation between clinical and non-clinical groups [56].

**Parenting Stress Index (PSI).** Parenting stress was measured using the Parenting Stress Index-Short Form (PSI-SF) [22], a 36-item self-report measure of parents' stress across three domains: 1) Parental Distress, 2) Parent-Child Dysfunctional Interactions, and 3) Difficult Child [23]. The scale has good internal consistency ($\alpha$ = .78 - .91) and test-retest reliability ($r$ = .61-.75) for subscale and total scale measures [23]. Parental distress was excluded from these analyses because it is largely overlapping with parent depression.

**Qualitative responses.** Of the total group of participants, $N$ = 551 and $N$ = 540 participants provided text-based responses to the following two questions: "What are the hardest things about parenting right now?" and "What are some of the things you wish you had that could help you with parenting right now?", respectively. Open-ended text responses were analyzed following thematic analysis, with the use of NVivo qualitative research software to assist with data organization [57, 58]. Thematic analysis followed the following stages: familiarization with data; line-by-line coding; development of larger meaning units for line-by-line codes; development of initial thematic framework-naming and defining themes and sub-themes; and review of thematic framework. Rigor–the quality, transparency, and thoroughness of qualitative analysis–was assured by documenting a detailed audit trail of the coding process and thematic framework development. NVivo hierarchy charts and maps were used to assess the representativeness of themes and sub-themes in the data. To enhance analytic trustworthiness, a second coder reviewed data and provided consensus in the interpretation of the thematic framework.

## Statistical analysis

Bivariate correlations between sociodemographic and household risk variables of interest were examined with parenting variables for inclusion in subsequent inferential analyses. A series of ordinary least squares (OLS) regressions in Mplus were used to predict each parenting variable from household risk factors and COVID-19 stressors (Table 3). We employed full information maximum likelihood estimation in Mplus to include all available data for participants with at least one dependent variable of interest. Post-hoc t-test analyses were used to examine the effect size of parenting differences between groups.

## Results

### Participants

Participants were predominantly mothers ($N$ = 568, 88.2%, $M$age = 35.37, $SD$ = 5.49, age range 21–66 years), married or common law, holding a bachelor's degree, and residing in Canada ($N$ = 570, 87% of total sample). The sample on average had high socioeconomic status; however, many were affected financially by the pandemic, including experiences of job loss, receipt government financial support, and difficulty managing unexpected expenses.34 Demographic

information is displayed in Table 1. This study was approved the University of Manitoba's Research Ethics Board.

## Quantitative results

Significant correlations were identified with household income, number of children, marital status, history of mental illness, relationship distress, employment loss, unmet childcare needs, and depression with at least one parenting variable (Table 2). Associated parenting variables included subscales on the PSI (i.e., Parent-Child Dysfunctional Interactions, and Difficult Child), The Parenting Scale (i.e., Laxness and Over-reactivity), and the PARYC (i.e., Supporting Positive Behavior, Setting Limits, and Proactive Parenting). Predictors not associated with any parenting variables were excluded from subsequent analyses. OLS regressions resulted in significant predictors of parenting variable for various household risk factors and COVID-19 stressors (Table 3). Across all parenting variables, caregivers with depression status reported riskier parenting, with effects sizes ranging from medium to large (Fig 1). Across all parenting variables, there were no significant differences in household COVID-19 risk factors between responders and non-responders.

## Qualitative responses

The uncontrollable and severe strains on parent capacities were highlighted across thematic domains (Table 4). Parents spoke about *challenges* in four domains including: Having too much time together; Self-doubt in parenting and teaching abilities; Role accommodation; and Managing COVID-19 psychological distress. Four main themes were evident for parenting *unmet needs* including: Childcare and additional supports; Resources and activities; More flexibility in work and school expectations; and Help managing psychological distress.

## Discussion

Here we provide emerging evidence for household factors and recent stressors linked to parenting quality during the COVID-19 pandemic. Parental depression status emerged as a strong and consistent predictor of lower-quality parenting (i.e., more stress, fewer positive behaviors, and more negative behaviors) across every indicator examined. Having multiple children in the home, unmet childcare needs, and relationship distress were additional risk factors for low-quality parenting across multiple indicators. Notably, these links remained after accounting for more stable household risk factors, including sociodemographic variables and history of parent mental illness. First-person qualitative accounts about the challenges of maintaining a positive parent-child relationship under pandemic conditions related to the four main themes of: Too much time together; Self-doubt in parenting and teaching abilities; Role accommodation; and Managing COVID-19 psychological distress; further underscore the impacts of COVID-19 to family functioning.

Common symptoms of depression include feelings of worthlessness and lower self-esteem and self-concept. Many participant responses reflected the theme of self-doubt, in their parenting abilities–in their abilities to connect with their children and to stimulate their growth intellectually, physically, emotionally, and socially. The link between quantitative and qualitative findings support a relationship where number of parenting stressors in addition to unmet needs (most lack of notably childcare, resources and activities, lack of flexibility with employment and home-schooling demands, and lack of psychological support) leave parents feeling overwhelmed and in doubt of their ability to manage their competing and challenging demands, thus further worsening depressive symptoms and increasing the impact on family functioning. These results build on recently published research on the impacts of COVID-19

**Table 1. Descriptive statistics.**

| Characteristic | No. (Valid %) | |
|---|---|---|
| **Stable Household Factors[a]** | | |
| Married or common law | 160 (43.4) | |
| Type of Caregiver | | |
| Mother | 568 (88.2) | |
| Other | 76 (11.8) | |
| Education[d] | | |
| Some or completed high school | 62 (9.5) | |
| Technical or Bachelor's degree | 339 (51.9) | |
| Professional, graduate degree or higher | 252 (38.5) | |
| Household income[e] | | |
| $0 - $60,000 | 102 (16.5) | |
| $60,001 - $120,000 | 257 (41.7) | |
| $120,001 + | 258 (41.8) | |
| **Proximal COVID-19 Factors[b]** | | |
| Mental Health History | 160 (43.4) | |
| Relationship distress (RDAS total) | 127 (37.2) | |
| Financial strain due to COVID-19 | 461 (70.9) | |
| Hours loss due to COVID-19 | 226 (36.7) | |
| Working < 50% normal hours due to COVID-19 | 59 (9.6) | |
| Food insecurity | 38 (10.2) | |
| Target child age | | |
| 18 months– 4 years | 451 (68.8) | |
| 5–8 years | 310 (47.3) | |
| Needs more childcare | 283 (44.4) | |
| **Parenting Domains[c]** | **_M (SD)_** | **No.** |
| The Parenting Scale (PS) | | |
| Laxness | 2.55 (.85) | 441 |
| Over-reactivity | 5.10 (.86) | 444 |
| Parenting Young Children (PARYC) | | |
| Supporting Positive Behaviours | 3.94 (.57) | 447 |
| Setting Limits | 3.69 (.52) | 445 |
| Proactive Parenting | 3.69 (.63) | 441 |
| Parenting Stress Index (PSI) | | |
| Parent-Child Dysfunctional Interaction | 19.94 (6.92) | 555 |
| Difficult Child | 29.70 (10.49) | 554 |

[a] Stable factors unlikely to change as a result of the pandemic.

[b] Factors likely impacted by the pandemic or contribute to increased parenting risk during the pandemic.

[c] measures of parenting quality across stress, positive, and negative parenting domains.

[d] Education was measured as a 7-level variable, with reductions in categories in the table for brevity.

[e] Household income was measured as a 16-level variable, with reductions in categories in the table for brevity.

on family well-being, including one study reporting high endorsement of psychological problems in older children in China [59] and another smaller sample of Italian families reporting higher than expected rates of parent distress [60]. Understanding specific factors linked to parenting quality in the relatively unprecedented context of a modern global pandemic is critical for guiding efforts to mitigate long-term child health and developmental risks.

**Table 2. Bivariate correlations.**

| | 1 | 2 | 3 | 4 | 5 | 6 | 7 | 8 | 9 | 10 | 11 | 12 | 13 | 14 | 15 | 16 | 17 | 18 |
|---|---|---|---|---|---|---|---|---|---|---|---|---|---|---|---|---|---|---|
| 1. Married | - | | | | | | | | | | | | | | | | | |
| 2. # of Children | .06 | - | | | | | | | | | | | | | | | | |
| 3. Annual Income | .34[c] | -.00 | - | | | | | | | | | | | | | | | |
| 4. Education | .08 | -.12[b] | .33[c] | - | | | | | | | | | | | | | | |
| 5. MH History | -.06 | .05 | -.14[a] | -.11[a] | - | | | | | | | | | | | | | |
| 6. Employment Loss | -.03 | .04 | -.28[c] | -.21[c] | .06 | - | | | | | | | | | | | | |
| 7. Financial Strain | -.09[a] | -.02 | -.31[c] | -.18[c] | .02 | .22[c] | - | | | | | | | | | | | |
| 8. Food Insecurity | -.14[b] | .02 | -.36[c] | -.20[c] | .10 | .18[b] | .22[c] | - | | | | | | | | | | |
| 9. Unmet Childcare Needs | -.05 | -.07 | .15[b] | .28[c] | .01 | -.14[b] | -.07 | .06 | - | | | | | | | | | |
| 10. Current Depression | -.04 | -.02 | -.14[b] | -.13[b] | .37[c] | .13[b] | .12[b] | .17[b] | .05 | - | | | | | | | | |
| 11. Relationship Distress | .01 | .05 | -.07 | -.08 | .17[b] | .05 | .06 | .13[a] | -.01 | .18[b] | - | | | | | | | |
| 12. Dysfunctional Int. | -.01 | .08 | -.09 | -.08 | .24[c] | .09[a] | .04 | .07 | .10[a] | .28[c] | .15[b] | - | | | | | | |
| 13. Difficult Child | .03 | .09[a] | -.02 | .02 | .30[c] | .06 | -.01 | .03 | .14[b] | .35[c] | .19[b] | .72[c] | - | | | | | |
| 14. Laxness | -.09 | -.09 | -.10[a] | -.01 | .16[b] | .04 | .09 | .10 | .08 | .22[c] | .05 | .22[c] | .21[c] | - | | | | |
| 15. Over-reactivity | -.06 | .12[a] | -.04 | .02 | .10 | .08 | .06 | .05 | -.02 | .23[c] | .12[a] | .45[c] | .46[c] | .30[c] | - | | | |
| 16. Supporting Positive | .11[a] | -.07 | .05 | -.09 | -.12[a] | -.05 | .01 | .03 | -.12[a] | -.12[c] | -.16[b] | -.44[c] | -.37[c] | -.18[c] | -.39[c] | - | | |
| 17. Setting Limits | .10[a] | -.01 | .08 | -.01 | -.09 | -.06 | -.05 | -.04 | -.11[a] | -.26[c] | -.18[b] | -.35[c] | -.33[c] | -.44[c] | -.45[c] | .65[c] | - | |
| 18. Proactive Parenting | .04 | -.06 | .07 | -.01 | -.06 | -.05 | -.04 | .04 | -.02 | -.19[c] | -.21[c] | -.29[c] | -.26[c] | -.23[c] | -.40[c] | .56[c] | .65[c] | - |

Abbreviations: MH = mental health; Int = interaction.

[a] $p < .05$.

[b] $p < .01$.

[c] $p < .001$.

Exposure to parent depression in early childhood can have a variety of harmful effects on child health and well-being [19, 61, 62]. Notably, however, depression is typically most impactful when it persists, is severe, and can be directly linked to changes in parenting behaviours [63, 64]. Given the recent onset of the pandemic, it is particularly concerning to observe a similar robust connection between parent depression and lower quality parenting in caregivers of young children. We expect that the pandemic context of household isolation is exacerbating the potential for parent depression to co-occur with parent-child relationship disruptions. At the same time, there are clear opportunities to limit intergenerational health risks given the growing body of research that indicates effectively treating both parental depression and parenting needs predicts improvements in child mental and physical health [10, 65]. Monitoring ongoing parent mental health and parenting needs, and intervening where appropriate, should be of high importance for public health efforts to promote child well-being.

One possible course of intervention is within the marital relationship given the possible indirect effects of poor relationship functioning on child development [66]. Marital or co-parent relationship distress is highlighted as a risk factor for a number of low-quality parenting factors, including dysfunctional parent-child interactions and reduced proactive parenting and limit setting. Inter-parental conflict is an established risk factor for the emergence of parent-child relationship disruptions as children observe and consequently learn to mimic emotionally dysregulated interactions between parents [67]. Spill-over effects can also occur in which parent distress leads to a reduced capacity to engage in more effortful high-quality parenting interactions, resulting in more negative parent-child relationships and associated negative health outcomes in children [30]. Similar to depression, we expect the impacts of relationship distress to have outsized effects on parent-child function during this time, given

**Table 3. Regression results.**

| | Parenting Stress | | | | Negative Parenting | | | | | | Positive Parenting | | | |
|---|---|---|---|---|---|---|---|---|---|---|---|---|---|---|
| | Dysfunctional Interactions | | Difficult Child | | Laxness | | Over-reactivity | | Proactive | | Supporting Positive Behaviors | | Setting Limits | |
| R2 (SE) | .19(.03) [c] | | .12(.03) [c] | | .07(.02) [b] | | .08(.03) [b] | | .08(.03) [b] | | .11(.03) [c] | | .10(.03) [c] | |
| | ß (SE) | 95% CI | ß (SE) | 95% CI | ß (SE) | 95% CI | ß (SE) | 95% CI | ß (SE) | 95% CI | ß (SE) | 95% CI | ß (SE) | 95% CI |
| **Family Factors** | | | | | | | | | | | | | | |
| Income | .03 (.12) | -.17 to .23 | -.08 (.08) | -.21 to .05 | -.01 (.01) | -.03 to .01 | .01 (.01) | -.01 to .03 | .00 (.01) | -.01 to .02 | .00 (.01) | -.02 to .01 | .00 (.01) | -.01 to .01 |
| Number of Children | 1.14 (.51) [a] | .30, 1.97 | .63 (.34) | .07 to 1.2- | -.08 (.05) | -.16 to .00 | .13 (.05) [a] | .05 to .21 | -.05 (.04) | -.12 to .01 | -.06 (.03) | -.11 to .01 | -.01 (.03) | -.06 to 03 |
| Mental Health History | 3.74 (1.08) [b] | 1.96 to 3.74 | 1.95 (.76) [a] | .71 to .3.19 | .14 (.10) | -.02 to .30 | .01 (.10) | -.15 to .17 | .06 (.07) | -.06 to .18 | -.01 (.06) | -.11 to .10 | .03 (.06) | -.07 to .13 |
| Marital Status | 1.60 (1.61) | -1.04 to 4.24 | .62 (1.07) | -1.14 to 2.38 | -.12 (.14) | -.34 to .11 | -.21 (.15) | -.44 to .03 | .07 (.12) | -.13 to .26 | .19 (.10) | .03 to .35 | .14 (.09) | -.01 to .30 |
| **Current Stressors** | | | | | | | | | | | | | | |
| Relationship Distress | 2.14 (1.10) | .33 to 3.95 | 1.12 (.76) | -.13 to 2.37 | -.01 (.10) | -.17 to .15 | .13 (.10) | -.03 to .30 | -.23 (.07) [b] | -.35 to -.11 | -.12 (.07) | -.22 to .01 | -.15 (.06) [a] | -.24 to -.05 |
| Employment Loss | 0.88 (.91) | -.62 to 2.38 | .82 (.62) | -.21 to 1.85 | .01 (.09) | -.13 to .16 | .07 (.09) | -.08 to .22 | -.01 (.07) | -.12 to .10 | -.04 (.06) | -.13 to .06 | -.03 (.05) | -.12 to .06 |
| Childcare needs | 2.91 (.86) [b] | 1.51 to 4.32 | 1.42 (.58) [a] | .46 to 2.38 | .11 (.08) | -.02 to .25 | -.04 (.08) | -.18 to .09 | -.02 (.06) | -.12 to .08 | -.12 (.05) [a] | -.21 to -.03 | -.10 (.05) [a] | -.18 to -.02 |
| Depression | 5.36 (.99) [c] | 3.73 to 6.98 | 2.65 (.67) [c] | 1.55 to 3.80 | .31 (.09) | .16 to .45 | .37 (.09) [c] | .22 to .52 | -.21 (0.7) [c] | -.32 to -.11 | -.27 (.06) [c] | -.37 to -.18 | -.25 (.05) [c] | -.34 to -.16 |

[a] p < .05

[b] p < .01

[c] p < .001

that young children are likely to be exposed to a higher proportion than usual while receiving less/no out-of-home care.

Unmet childcare needs also emerge as predictive of lower-quality parenting including elevated dysfunctional parent-child interactions, perceptions of difficult child temperament, and reductions in supporting positive behavior and setting limits. Having a higher number of children in the home was also associated with greater over-reactive parenting, which is a particular concern because it is a risk factor for harsh parenting and maltreatment [68]. Due to mandatory reporting requirements, more severe forms of discipline were not explored here but should be examined in subsequent research that is better equipped to respond effectively to address identified child safety risks.

Notably, these associations between psychosocial risks and low-quality parenting are emerging in a relatively high SES sample compared to the general public (e.g., 2018 median annual family income in Canada = $87,930 [69]), as is common for large-scale online recruitment techniques. The fact that these risks emerge with such strength here is concerning, given that the sample is less likely than the general population to experience co-occurring stressors (e.g. poverty) that further impact parents' capacity to provide the rich responsive caregiving established to promote healthy development [10]. However, we found that the additional stressors of employment loss or financial strain during the pandemic were not significantly associated with parenting in this sample, despite robust reports of parenting risk associated with economic hardship or recession [70, 71]. It is possible that the economic impacts of these

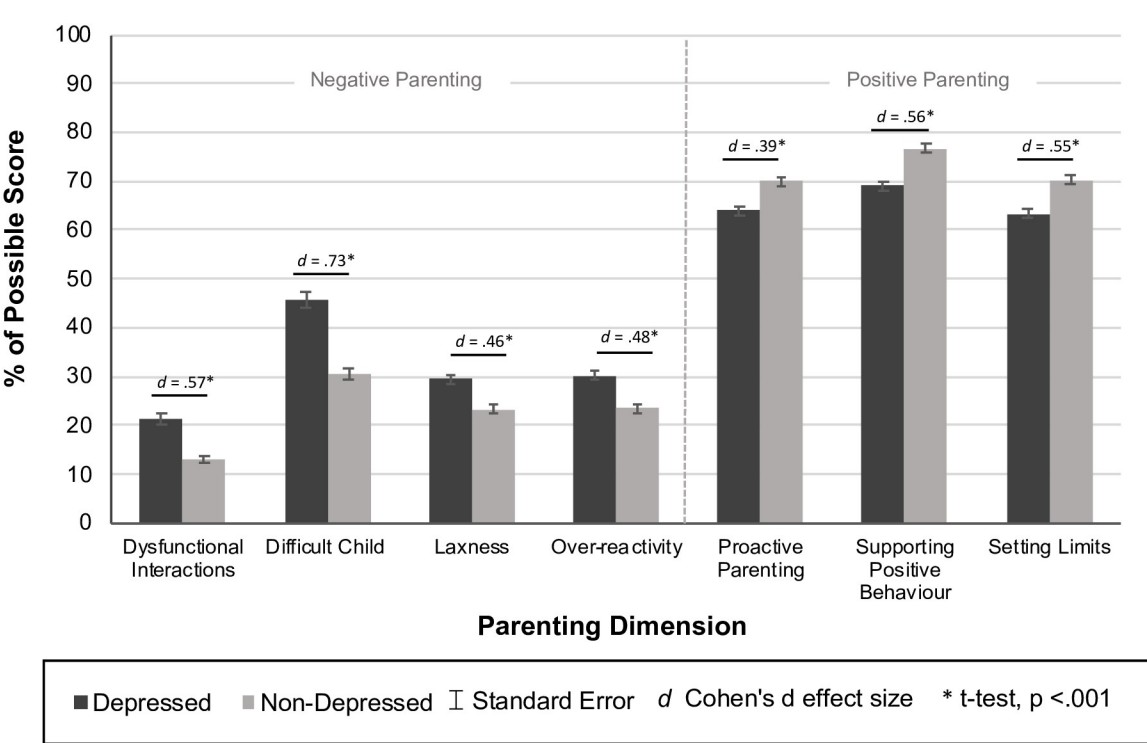

**Fig 1. Positive and negative parenting in depressed and non-depressed caregivers during COVID-19.**

factors in our overall high SES samples has less impact on parenting behaviour than would be the case for families with already low SES. In addition, our questionnaire was collected in the first few months of the pandemic in which it's possible that the longer-term effects of economic hardships were not reflected in parenting behaviors. However, our online survey also allowed for participant anonymity, as providing contact information was optional. Anonymity on self-report measures of highly stigmatized behaviours, including child maltreatment, has been found to lead to greater likelihood of behaviour disclosures [72]. Given the stigma associated with low-quality parenting, our results could have more ecological validity due to survey anonymity. of household and COVID-19 risk factors on parent functioning. Nevertheless, we provide a timely snapshot of proximal pandemic and stress-related factors linked to greater parenting risk. As data were collected across jurisdictions differentially affected by the pandemic regarding illness, economic burden, and lock-down status, the identified risk factors for lower-quality parenting may vary across jurisdictions with different family and community needs.

Our sample also included a majority of female participants (i.e., mothers), consistent with other research from convenience samples, which limits generalizability of findings to other parenting roles [73]. However, emerging research highlights a gender division of childcare and parenting responsibilities that has disproportionately affected mothers during the pandemic [71]. Given that the majority of child-care tasks are often carried by mothers, it is not unexpected to have mothers responding at a disproportionately higher rate because of their primary role in parenting [74]. This gender gap has implications for how parenting quality of mothers

**Table 4. Qualitative responses-parenting challenges and needs.**

| Main Themes | Description | Quotes |
|---|---|---|
| **Parenting Challenges** | | |
| "Too much time together" | "Too much time together" with children was described as one of the hardest things about being a parent during this pandemic. Participants noted the strain that this lack of time creates on mood and on marital relationships, in addition to the impact on energy and engagement in parenting. | "Spending 24 hours a day with each other with no break in sight" |
| | | "Trying to give so much time to both kids leaves us stretched thin" |
| | | "Never having a mental break from my kids until bedtime" |
| | | "No time to turn my brain off, on alert all the time, it's exhausting" |
| | | "No mental breaks. [I] hide myself in stairwell or shower for alone time" |
| | | "We never get a break, never get time alone together, so no time to reenergize" |
| | | "Zero time to ourselves as parents, no privacy to express our difficult emotions without kids being exposed to that. Sticking with schedule when we feel unmotivated, exhausted, depressed. Kids going to bed way too late, so we have no time to ourselves at all—marital relationship suffering." |
| "Am I doing enough for my child" | Participants expressed concern over whether they were engaging their children in sufficient activities to stimulate their cognitive, physical, and social development. Participants with multiple children noted a particular challenge of finding activities that would suit different learning levels and needs. The loss of regular family and peer contact as well as closure of parks and playgrounds due to social (physical) distancing were noted as barriers to cognitive and physical stimulation | "Feeling like I'm pretty crap at everything–I can't do any of it justice" |
| | | "Not having our regular outing activities such as playgroups, libraries, play places, restaurants, or even being able to take the kids grocery shopping with me. Also, not having other parents and their kids over which provided social stimulation for both the adults and the kids" |
| | | "Making sure he is getting enough exercise (he was a very active kid before this- hockey ended early, swimming lessons and baseball for the spring are cancelled). Limiting screen time is a challenge." |
| | | "Trying to be the perfect parent and use every moment as a teaching moment, feeling bad when my children are using more screen time than normal" |
| | | "Trying to keep them entertained and stimulated, making sure they get enough physical activity, making sure I have the patience" |
| | | "The hardest thing is feeling like I'm letting my own kids down. We are sent lessons but there is no virtual class instruction or video lessons for my older child to learn from before he begins the assignment, and they are all in French. We end up having to interpret and teach the pre-lesson so that he can do the work more independently, but it's slow" |
| | | "There's a crushing responsibility to keep your kids safe always, but's it's really amplified now. It's hard to be calm and regulated all the time to keep the kids stress lower" |
| "Too many roles!" | Participants outlined their distress surrounding the number of roles and responsibilities they are currently being asked to balance simultaneously, for many hours of the day, and the feeling of "juggling these roles" | "Keeping the role of a worker, teacher, childcare, housekeeper and wife is not easy" |
| | | "Impossible to maintain expected work productivity with a toddler at home" |
| | | "Trying to work and parent makes it hard to do either well" |
| | | "The most challenging part has been working through the amount of schoolwork the kids have and trying to create a learning environment with the activity of home life constantly in the background" |
| | | "Trying to parent while both parents are working from home. Feeling a lot of guilt around how I'm parenting, the work or lack of work I'm getting done, trying to balance but feeling like I'm failing at both" |
| | | "Working from home, parenting, school time, managing emotional responses, keeping household running all collide at the same time with no means of escape" |
| | | "Feeling like I'm doing a terrible job of everything. Feeling scattered all the time, like juggling a dozen balls all day long and dropping most of them. Feeling half connected to everything; Terrible difficulty focusing on work and constantly pulled away from one thing to another" |

(*Continued*)

**Table 4.** (Continued)

| Main Themes | Description | Quotes |
|---|---|---|
| Managing parent- and child-COVID-19- related psychological distress | Parents described challenges coping with their own distress and mental health and the mental health of their children during this uncertain time (stress, guilt, frustration, worry, anxiety, sadness). Parents also described concern over how to discuss COVID-19 with their children | "I'm worried about my imminent and future employment security. I cannot function well as a parent in these conditions and find I become easily angered and frustrated with them. Then feel guilty" |
| | | "Managing the extreme emotions (made more extreme by the changes in circumstances), the lack of alternate child entertainment (friends, parks), increased nightmares (and thus decreased sleep for everyone). It goes on" |
| | | "I am more irritable and less patient with my son (and husband) than I used to be because I am so stressed and over-tired" |
| | | "Worrying about the long-term impact this could have on our kids" |
| | | "Constant fighting, kids are anxious, managing their emotions (they miss friends, miss structure)" |
| | | "The hardest thing is seeing my daughter's distress-she is acting out and we think it is because she can't comprehend why things aren't normal" |
| | | "Explaining to them [kids] that they can't go to school or see grandparents and family" |
| | | "Communicating with young children effectively about COVID-19" |
| **Parenting Needs** | | |
| Childcare and additional supports | Childcare was the most prominent need reported across participant responses. Participants reported direct childcare needs, as well as other supports, including assistance with teaching and other areas of social, cognitive, and physical development in children | "More childcare and/or more active support from teachers" |
| | | "Childcare facility providing activities and ideas to try with my toddler" |
| | | "An extra hand for even 30 minutes to an hour a day to watch/play with the children while I focus on work" |
| | | "I so wish childcare centers were still open. That is the only thing that would make this situation manageable" |
| | | "More formal teaching provided in a schedule from school" |
| | | "More video lessons" |
| | | "I'd love help with teaching my school aged children. I'm not a teacher and to pick things up ¾'s of the way through the school year is difficult" |
| Resources and activities | Participants reported a need for resources and activities that could assist with the cognitive, physical, emotional, and social developmental needs of their children | "More arts and crafts supplies, more activity ideas, help with the children so I can take a short break for some self-care" |
| | | "Ways to burn energy. We are limited in going outside because we have an immune compromised child" |
| | | "There are a ton of resources out there but they are scattered and many not suitable for pre-reading children. Something to address this would be helpful" |
| | | "I wish I had more space for the kids to romp around and play, more board games, books, art materials, etc" |
| | | "More access to activities, crafts, emotional support" |
| More flexibility in expectations needed from the workplace and from schools | Participants described a need for more flexibility from both the workplace and from schools in balancing the many roles that they are playing simultaneously | "More flexibility from work in terms of hours required per week so more time can be dedicated to giving attention to our boys" |
| | | "More flexibility from schools for littles–let them learn through play and stop asking us to do the impossible" |
| | | "Reduced expectations! Some level of certainty around teaching and learning expectations" |
| | | "It would help so much if work would acknowledge that no one can be super productive now, especially with young children and no childcare. If work expectations were adjusted I would be less stressed and more emotionally present with my kids" |
| | | "Definitely more empathy, support and compassion from my employer in recognizing I am in a position where I am a parent needing to be working from home without childcare or a spouse who is able to be home and help. It is an impossible job to work full time and provide meaningful learning and care to a preschooler. Both my child and my own mental health has suffered greatly by the added pressures/ expectations put on by my employer in such a difficult time" |

(*Continued*)

**Table 4.** (Continued)

| Main Themes | Description | Quotes |
|---|---|---|
| Help managing psychological distress in parents and children | Parents noted that they need help managing psychological distress, both for themselves and their children | "Mental health services—for myself in coping and for my child. I realize that these services are 'available' but typically have very long waiting lists, or are prohibitively expensive" |
| | | "We need mental health support" |
| | | "Stress reduction strategies to help with better sleep" |
| | | "I have tried meditating off and on, most of the tricks do not work" |
| | | "Online video support group or chat group in evenings" |
| | | "Connection with other moms" |
| | | "Typical coping strategies no longer practiced due to COVID-19 (e.g., gym, shopping)" |
| | | "Managing uncertainty" |
| | | "Self-care" |
| | | "There is no specific guidance for managing toddlers and their well-being through this crisis" |
| | | "Age-appropriate books for kids to read about how to deal with stress/uncertainty of life at the moment" |
| | | "Access to counselling for our oldest child to help with anxiety" |

and fathers may be differentially impacted by psychosocial and pandemic-related stressors. A final methodological limitation was the use of self-reports of mental health which capture symptom levels, not clinical diagnoses [75]. However, self-report of mental health is widely used in public-health observational research and have been shown to have high sensitivity and specificity for diagnosis [76–78].

## Conclusion

Taken together, numerous pandemic-linked risk factors are identified as significant predictors of parenting stress and behaviours. These are the first data examining individual differences in household factors and pandemic-link stressors to parenting quality during COVID- 19, which is a critical step to targeting health services responses at practice and policy levels.

High quality parenting and positive relationships are understood to be critical for helping young children cope with stressors of varying severity. Over the past decade, numerous calls by pediatric leaders, such as the World Health Organization (WHO), highlight the critical nature of providing mental health and parenting support to primary caregivers of young children [10, 19]. In the absence of targeted action to support caregiver mental health, our data suggest that young children may be exposed to lower-quality parenting and disruptions in family relationships during the COVID-19 pandemic, with potential for significant long-term health and developmental risks.

Next steps for policy makers and health systems aiming to protect child health and development include the dissemination of mental health support for parents. In order to address the widespread unmet parent mental health needs during the pandemic, key considerations will be critical including: implementing evidence-based programs that can treat both depressive symptoms and promote supportive parenting, ensuring supports are low cost, considering a variety of online, telephone, or physically distanced service delivery options to accommodate family schedules and comply with physical distancing public health recommendations.

Novel technologies providing digital delivery of psychological services for families are playing a crucial role during the pandemic [79–81], when in-person care has become less

accessible. Particularly for families experiencing a range of household and psychosocial risk factors, providing evidenced-based telehealth services is critical for preventing coercive parent-child interactions and potentially more severe outcomes such as child abuse and maltreatment. Current digital parenting interventions hold promise for improving a range of family outcomes including parenting skills, parent and child mental health, and parenting stress [78–81]. Telehealth programs such as Triple P Online [82] and ezPARENT [83] are documented to have similar effect sizes to other evidenced-based parenting programs offered in-person [78] but may be able to maximize service accessibility and address family health needs now.

Unmet childcare and associated stressors are also a prominent need arising from quantitative and qualitative findings. Qualitative responses indicated that parents are experience distress associated with competing and challenging roles in addition to trying to accommodate role-related responsibilities in the face of scare supports. Further investments in safe and low-costs childcare or access to resources and activities are needed to reduce parent distress and the associated child harms that can arise from harsh or unsupportive parenting. As we globally move towards a second wave of COVID19, young children at home are expected to be highly vulnerable. Emerging research has shown substantial drops in reports of child maltreatment during the pandemic, presumably because professionals (e.g., educators, care providers) are spending significantly less in person time with children due to physical distancing mandates [84]. At the same time, longitudinal studies have indicated increased parent-child conflict, abuse risk, and psychological aggression [85]. Greater consideration of parenting stress and mental health in particular is necessary to limit children's risk at home and prevent significant harm. Increased vigilance by educator, physician, and other professionals in online and in person settings as well as the provision of timely, evidenced-based supports will be crucial for reducing widespread child developmental risks that is linked to elevated levels of parent stress and mental health problems.

This study provides timely evidence for potential household and pandemic-related risk factors that may contribute to increase parental stress and diminished parenting quality during the COVID-19 pandemic. Given ongoing government-mandated restrictions and associated impacts on families that are likely to persist (e.g., unstable childcare and schooling, decreased provision of family resources, financial strain and job insecurity), our findings highlight the importance of prioritizing the needs of families in order to prevent long-term psychological impacts. Even following the pandemic, potential harms to parent-child relationships and chronic stress exposure incurred during lockdowns may have negative long-term developmental consequences for young children. In response to the pandemic, we recommend careful monitoring of child and parent well-being by health care providers and teachers alongside greater access to evidence-based mental health and parenting supports to promote resiliency.

## Author Contributions

**Writing – original draft:** Marlee Salisbury, Lara Penner-Goeke, Emily E. Cameron.

**Writing – review & editing:** Leslie E. Roos, Jennifer L. P. Protudjer, Ryan Giuliano, Tracie O. Afifi, Kristin Reynolds.

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
