## [Decision Letter · Decision Letter 0]

29 Jan 2021

PONE-D-20-29512

Supporting families to protect child health:

Parenting quality and household needs during the COVID-19 pandemic

PLOS ONE

Dear Dr. Roos,

Thank you for submitting your manuscript to PLOS ONE. After careful consideration, we feel that it has merit but does not fully meet PLOS ONE’s publication criteria as it currently stands. Therefore, we invite you to submit a revised version of the manuscript that addresses the points raised during the review process.

I agree with the reviewers' comments that this study is very interesting and dealing with one of the important issues of COVID-19. I expect that the reviewers' suggestion would greatly increase the impact of this paper.

We look forward to receiving your revised manuscript.

Kind regards,

Kyoung-Sae Na, M.D.

Academic Editor

PLOS ONE

2. We noted in your submission details that a portion of your manuscript may have been presented or published elsewhere. ["Results on the prevalence of maternal mental health have been published elsewhere (Journal of Affective Disorders)."] Please clarify whether this publication was peer-reviewed and formally published. If this work was previously peer-reviewed and published, in the cover letter please provide the reason that this work does not constitute dual publication and should be included in the current manuscript.

Reviewers' comments:

Reviewer's Responses to Questions

**Comments to the Author**

1. Is the manuscript technically sound, and do the data support the conclusions?

Reviewer #1: Yes

Reviewer #2: Yes

Reviewer #3: Yes

2. Has the statistical analysis been performed appropriately and rigorously? 

Reviewer #1: Yes

Reviewer #2: Yes

Reviewer #3: Yes

3. Have the authors made all data underlying the findings in their manuscript fully available?

Reviewer #1: No

Reviewer #2: Yes

Reviewer #3: Yes

4. Is the manuscript presented in an intelligible fashion and written in standard English?

Reviewer #1: Yes

Reviewer #2: Yes

Reviewer #3: Yes

5. Review Comments to the Author

Reviewer #1: This is a very interesting paper regarding parenting and parental mental health during Covid-19 in a cross sectional online survey. A great advantage is that the authors have used several interesting validated questionnaires and add qualitative data. The introduction is well written. However, I do have some concerns that should be clarified:

• Could you add psychometrics of the used questionnaires?

• You have assessed academic background, which is known to be highly important for parenting and stress coping. Why haven’t you included this important factor into regression analyses

• Same for age and gender of parents and children – this could be important family characteristics and predict parenting

• You mention SES as important factor and that usually, sample in online surveys have a higher SES compared to the general public. To get a better idea on the representativeness of your sample, it would be good to have a comparison with e.g. SES of the general public of parents in Canada

• It is surprising that based on all reports on economic hardship and parenting during times of e.g. recession, this was not a significant predictor of parenting. Could you discuss this?

• Limitations section: Majority of participants were females – please discuss how this can decrease the external validity of the study

Reviewer #2: It’s a very interesting and important work! The paper is well written with key points made easy to find. The methods and study design are clear, thoroughly described, and well-matched to answer the stated research questions. Here are some concerns:

Introduction

“Families are coping with numerous challenges including unmet childcare needs, crisis schooling, low resource access, and financial strain.” Citations are needed. Also, may consider including the impact of social distancing and quarantine. For example, authors may find evidence and strategies from some recent articles:

Janssen, Loes HC, Marie-Louise J. Kullberg, Bart Verkuil, Noa van Zwieten, Mirjam CM Wever, Lisanne AEM van Houtum, Wilma GM Wentholt, and Bernet M. Elzinga. "Does the COVID-19 pandemic impact parents’ and adolescents’ well-being? An EMA-study on daily affect and parenting." PloS one 15, no. 10 (2020): e0240962.

Ye J. Pediatric mental and behavioral health in the period of quarantine and social distancing with COVID-19. JMIR pediatrics and parenting. 2020;3(2):e19867.

Methods

It’s not clear why the authors chose 1.5-8 years old children for this study. Any evidence for this range?

Did this study get IRB approval? If so, please include this information in the main text.

There was missing data based on the results Table 1. The missing rates in some variables were pretty high. How did you handle missing data in your statistical analyses?

How to interpret Table 2 in terms of its relationship with your primary aim? More explanations are needed to help the readers understand this table.

Results

It will be helpful to link the quantitative and qualitative results, and give some feasible interventions from the family-level based on your findings. The interventions that authors mentioned in the discussion may not be generalizable in other countries or settings.

Reviewer #3: dear Author I Congratulate you on the submission, the work is good and well conducted. the tools used are appropriate. I would like to point out to the authors that there can be a section that gives the teachers and professionals involved in managing the children a recommendation to manage and anticipate the needs post the COVID-19 Crisis thus enabling them better management of the children's need.

6. PLOS authors have the option to publish the peer review history of their article (what does this mean?). If published, this will include your full peer review and any attached files.

Reviewer #1: No

Reviewer #2: No

Reviewer #3: **Yes: **Deblina Roy

---

## [Author Response · Author response to Decision Letter 0]

19 Mar 2021

-- We have reviewed the style templates and adjusted the manuscript.

2. We noted in your submission details that a portion of your manuscript may have been presented or published elsewhere. ["Results on the prevalence of maternal mental health have been published elsewhere (Journal of Affective Disorders)."] Please clarify whether this publication was peer-reviewed and formally published. If this work was previously peer-reviewed and published, in the cover letter please provide the reason that this work does not constitute dual publication and should be included in the current manuscript.

-- This is the second publication from a large online convenience sample. The current study highlights novel findings relevant to factors associated with parenting quality, while the previously published solely examined maternal mental health status and protective factors.

-- No changes at this time, we will update Data Availability upon acceptance

Response to Reviewers

Reviewer #1: This is a very interesting paper regarding parenting and parental mental health during Covid-19 in a cross sectional online survey. A great advantage is that the authors have used several interesting validated questionnaires and add qualitative data. The introduction is well written. However, I do have some concerns that should be clarified:

• Could you add psychometrics of the used questionnaires?

-- We have ensured that psychometrics are included for each questionnaire used in the methods (pp. 8-9).

• You have assessed academic background, which is known to be highly important for parenting and stress coping. Why haven’t you included this important factor into regression analyses

-- We appreciate the attention to the relevance of parent education to parenting stress and coping. However, bivariate correlations (Table 2.) indicated that parent education was not associated with any of the parenting measures in our sample, so was not brought into multivariate analyses, consistent with our analytic plan. 

• Same for age and gender of parents and children – this could be important family characteristics and predict parenting

-- We agree that child age and gender would be important predictors of parenting to include in analyses. However, the survey design did not allow for data on these specific demographics for the child considered in parent’s responses. Parents reported gender and age of all children but were advised to only think about their most challenging child when responding to relevant questionnaires. Because the majority of families reported having more than one child, we are unable to determine the exact age and gender of each child considered in the parenting questionnaires responses. This procedure is described on pp 6-7.

• You mention SES as important factor and that usually, sample in online surveys have a higher SES compared to the general public. To get a better idea on the representativeness of your sample, it would be good to have a comparison with e.g. SES of the general public of parents in Canada

-- We have now included 2018 census data describing the median annual family income in Canada as a comparison to our sample (p. 25).

• It is surprising that based on all reports on economic hardship and parenting during times of e.g. recession, this was not a significant predictor of parenting. Could you discuss this?

-- Although economic hardship has been previously reported to impact parenting, it is possible we did not find this effect because of our relatively high SES sample in which financial strain during the pandemic may be less impactful. We have now discussed this further on pp. 25-26. 

• Limitations section: Majority of participants were females – please discuss how this can decrease the external validity of the study

-- Thank you for pointing out this limitation of our sample. We have provided a discussion of this limitation and its relevance to family life on p.26. 

Reviewer #2: It’s a very interesting and important work! The paper is well written with key points made easy to find. The methods and study design are clear, thoroughly described, and well-matched to answer the stated research questions. Here are some concerns:

Introduction

“Families are coping with numerous challenges including unmet childcare needs, crisis schooling, low resource access, and financial strain.” Citations are needed. Also, may consider including the impact of social distancing and quarantine. For example, authors may find evidence and strategies from some recent articles:

Janssen, Loes HC, Marie-Louise J. Kullberg, Bart Verkuil, Noa van Zwieten, Mirjam CM Wever, Lisanne AEM van Houtum, Wilma GM Wentholt, and Bernet M. Elzinga. "Does the COVID-19 pandemic impact parents’ and adolescents’ well-being? An EMA-study on daily affect and parenting." PloS one 15, no. 10 (2020): e0240962.

Ye J. Pediatric mental and behavioral health in the period of quarantine and social distancing with COVID-19. JMIR pediatrics and parenting. 2020;3(2):e19867.

-- Thank you for pointing out our oversight in referencing the relevant literature. We have now included three relevant citations pertinent to our child age group that support this statement on p.3, including the Ye (2020) article. 

Methods

It’s not clear why the authors chose 1.5-8 years old children for this study. Any evidence for this range?

-- We have provided justification for using this age range on pp 7-8. Briefly, we wanted to distinguish our sample from the postpartum period, when parent-child interactions may differ from those during early childhood. As well, early childhood is generally considered to be until age 8, during which children are highly susceptible to environment influence including parenting factors. Citations are included to support this decision-making.

Did this study get IRB approval? If so, please include this information in the main text.

-- Yes it did. This is now included in the main text (p.11)

There was missing data based on the results Table 1. The missing rates in some variables were pretty high. How did you handle missing data in your statistical analyses?

-- Missing data was handled in Mplus with Maximum Likelihood approaches. We have now described this procedure on p. 10. 

How to interpret Table 2 in terms of its relationship with your primary aim? More explanations are needed to help the readers understand this table.

-- We have now more clearly explained the variables included in Table 2 in the results section (pg. 12).

Results

It will be helpful to link the quantitative and qualitative results, and give some feasible interventions from the family-level based on your findings. The interventions that authors mentioned in the discussion may not be generalizable in other countries or settings.

-- Thank you for this recommendation to provide more discussion on effective evidenced-based interventions that can support families during the pandemic. We have included this in our discussion on p.28.

Reviewer #3: dear Author I Congratulate you on the submission, the work is good and well conducted. the tools used are appropriate. I would like to point out to the authors that there can be a section that gives the teachers and professionals involved in managing the children a recommendation to manage and anticipate the needs post the COVID-19 Crisis thus enabling them better management of the children's need.

-- Thank you for this suggestion. We have now included some recommendations in consideration of educators and clinical professionals on p.28-29.

---

## [Decision Letter · Decision Letter 1]

6 Apr 2021

PONE-D-20-29512R1

Supporting families to protect child health:

Parenting quality and household needs during the COVID-19 pandemic

PLOS ONE

Dear Dr. Roos,

Thank you for submitting your manuscript to PLOS ONE. After careful consideration, we feel that it has merit but does not fully meet PLOS ONE’s publication criteria as it currently stands. Therefore, we invite you to submit a revised version of the manuscript that addresses the points raised during the review process.

The reviewer raised one minor issue.

We look forward to receiving your revised manuscript.

Kind regards,

Kyoung-Sae Na, M.D.

Academic Editor

PLOS ONE

Journal Requirements:

Reviewers' comments:

Reviewer's Responses to Questions

**Comments to the Author**

1. If the authors have adequately addressed your comments raised in a previous round of review and you feel that this manuscript is now acceptable for publication, you may indicate that here to bypass the “Comments to the Author” section, enter your conflict of interest statement in the “Confidential to Editor” section, and submit your "Accept" recommendation.

Reviewer #2: (No Response)

2. Is the manuscript technically sound, and do the data support the conclusions?

Reviewer #2: (No Response)

3. Has the statistical analysis been performed appropriately and rigorously? 

Reviewer #2: (No Response)

4. Have the authors made all data underlying the findings in their manuscript fully available?

Reviewer #2: (No Response)

5. Is the manuscript presented in an intelligible fashion and written in standard English?

Reviewer #2: (No Response)

6. Review Comments to the Author

Reviewer #2: Thank you for addressing the comments. One additional suggestion is to add a conclusion section and summarize your key findings and recommendations.

7. PLOS authors have the option to publish the peer review history of their article (what does this mean?). If published, this will include your full peer review and any attached files.

Reviewer #2: No

---

## [Author Response · Author response to Decision Letter 1]

12 Apr 2021

In response to reviewer comments,

“Thank you for addressing the comments. One additional suggestion is to add a conclusion section and summarize your key findings and recommendations.”

we have organized a conclusion section and summarized our findings to provide concluding recommendations for supporting families of young children during and after the COVID-19 pandemic.

---

## [Editor Report · Decision Letter 2]

3 May 2021

Supporting families to protect child health:

Parenting quality and household needs during the COVID-19 pandemic

PONE-D-20-29512R2

Dear Dr. Roos,

We’re pleased to inform you that your manuscript has been judged scientifically suitable for publication and will be formally accepted for publication once it meets all outstanding technical requirements.

Kind regards,

Kyoung-Sae Na, M.D.

Academic Editor

PLOS ONE
---

## [Editor Report · Acceptance letter]

14 May 2021

PONE-D-20-29512R2 

Supporting families to protect child health:
Parenting quality and household needs during the COVID-19 pandemic 

Dear Dr. Roos:

I'm pleased to inform you that your manuscript has been deemed suitable for publication in PLOS ONE. Congratulations! Your manuscript is now with our production department. 

Kind regards, 

on behalf of

Dr. Kyoung-Sae Na 

Academic Editor

PLOS ONE